# The Study on Spatial Elements of Health-Supportive Environment in Residential Streets Promoting Residents’ Walking Trips

**DOI:** 10.3390/ijerph17145198

**Published:** 2020-07-18

**Authors:** Shaohua Tan, Fengxiao Cao, Jinsu Yang

**Affiliations:** School of Architecture and Urban Planning, Chongqing University, Chongqing 400030, China; tsh@cqu.edu.cn (S.T.); 20081502106@cqu.edu.cn (J.Y.)

**Keywords:** residential streets, health-supportive environment, residents’ walking trips, spatial elements, spatiotemporal characteristics, health needs

## Abstract

Residents’ walking trips are a kind of natural motion that promotes health and wellbeing by modifying individual behavior. The purpose of this study was to evaluate the major influence of the spatial elements of a health-supportive environment on residents’ walking trips. This study analyzes residents’ walking trips’ elements based on the spatiotemporal characteristics of walking trips, as well as the spatial elements of a health-supportive environment in residential streets based on residential health needs. To obtain the spatial elements that promote residents’ walking trips and to build an ordered logistic regression model, two methods—a correlation analysis and a logistic regression analysis—were applied to analyze the elements of residents’ walking trips as well as the spatial elements of a health-supportive environment in residential streets by means of SPSS software, using on-site survey results of ten residential streets and 2738 pieces of research data. The research showed that the nine kinds of spatial elements that significantly affect residents’ walking trips are density of pedestrian access, density of bus routes, near-line rate of roadside buildings, average pedestrian access distance, square area within a 500 m walking distance, distance to the nearest garden, green shade ratio, density of street intersections, and the mixed proportion of differently aged residential buildings. In order to construct a spatial environment that promotes walking trips, it is necessary to improve the convenience of residents’ walking trips, to increase the safety of roadside buildings and pedestrian access, to expand the comfort of “getting out to the nature”, and to enrich the diversity of different architectural styles and street density.

## 1. Introduction

Physical activity (PA) is a kind of generally recognized protective factor. However, physical inactivity (PI) has become a universal problem around world [1]. According to statistics, 31.1% of the world’s adults cannot reach the standard level of physical activity. The China National Health and Planning Commission (CNHPC) found that the regular exercise rate of adults is only 18.7% in China [2]. Many studies have shown that physical inactivity (PI) has a substantial influence on obesity, heart disease, hypertension and diabetes, among others. It has become the most serious challenge of global public health in the 21st century [3,4,5,6,7]. The question of how to scientifically and reasonably intervene and improve the physical activity of residents is particularly important for improving the physical health of the general population; it is also of particular importance for the “Healthy China” (HC) strategy [8,9]. Improving the health level of residents via environmental interventions is the key aim of Healthy City Construction (HCC) in China. Constructing a health-supportive environment is one of the important goals of healthy city construction (The “Medium and long term plan for chronic disease control in China (2017–2025)” issued by the general office of the State Council of China has stated that controlling risk factors, building a health-supportive environment and setting up a series of health-supportive environment construction projects are one of the important goals of healthy city construction). The residential area, as the basic unit of the city, is the basic object of a health-supportive environment [10].

### 1.1. Residents’ Walking Trips

Residents’ walking trips are a widely endorsed way to increase physical activities [11,12,13,14,15,16]. Residents’ walking trips are an important component of outdoor activities, and are an accessible, low-risk and inexpensive physical exercise style that can prevent and treat non-communicable diseases [17,18,19], such as cardiovascular disease and obesity, and improve overall health [20]. Many researches in China and abroad explored the impact of residents’ walking trips on residential health, and it is considered that residents’ walking trips are one of the most convenient and economical exercise modes [21]; additionally, they are beneficial to physical health and mental pleasure. There are many studies on residents’ walking trips for differently aged and genders [22,23], but few studies on residents’ walking trips based on time and space characteristics. The elements of residents’ walking trips were analyzed and studied based on the spatiotemporal characteristics of residents’ walking trips in this study.

### 1.2. Health-Supportive Environment in Residential Streets

The World Health Organization (WHO) defines a health-supportive environment as an autonomous environment that is provided to the public free from health threats and that enables them to develop their capabilities and health [24]. A health-supportive environment in residential streets is the one that can meet residential health needs, which include physical health needs, psychological health needs, social adaptability health needs, and moral health needs [25]. A health-supportive environment in residential streets plays a key role in improving residential physical activities (PA). Physical activities (PA) have been integrated into daily residential life by encouraging residents’ walking trips [26].

In 2015, the healthy places design guidelines issued by the American Urban Land Institute proposed 17 planning and designing tools, including a mix of urban functions, street density, building density, slow traffic systems and natural environment accessibility, among others, which have a great impact on urban planning in North America [27]. Chinese scholars believe that the improvement of the constructed environment is an important starting point for urban planning to actively intervene in public health. The focus in improving residential health in residential streets is on constructing space for residential physical activities and improving residential participation in residential streets [28]. One of the sustainable methods that can encourage resident to increase physical activities is building a spatial environment in residential streets that promotes residents’ walking trips [29,30]. On the basis of the above research, starting from the residential health needs and combining with the current situation of the spatial environment in residential streets, the spatial elements of a health-supportive environment in residential streets are analyzed in this study.

### 1.3. Research Contents

This paper mainly studies the influence of spatial elements of a health-supportive environment in residential streets on residents’ walking trips. The main research contents are as follows:

Elements of residents’ walking trips in residential streets;

Spatial elements of a health-supportive environment in residential streets;

Spatial elements of a health-supportive environment in residential streets that promote residents’ walking trips.

## 2. Materials and Methods

### 2.1. Research Area

The Nan’an District, in Chongqing City, was selected as the research area for this study. The Nan’an District is the core area in Chongqing city, covering an area of 262.43 square kilometers, with a resident population of 530,000 people. It is surrounded by the Yangtze River to the north and west. The overall terrain is relatively flat, with a few gentle slopes, as is common in mountainous cities. To avoid an impact of the business district on residents’ walking trips, ten residential streets within a straight-line distance of more than 1 km from Nanping business district were selected as the research samples (Figure 1). The research area’s scope is the street space within a 500 m walking path from the sampled residential area. At present, most of the residential areas in China are access-controlled residential areas with walls, and residents and visitors must be authenticated before entering the residential area. In Figure 1, the yellow area is the range of the residential area, the green dot is the pedestrian access of the residential area, and the red line represents the 500 m walking path starting from the pedestrian access of the residential area.

Residents’ walking trips’ data from the sample residential areas were gathered using on-site surveys and questionnaires. Therefore, there was a large amount of data on residents’ walking trips in one residential area. Spatial elements’ data from the sampled residential areas were gathered using an on-site survey, street view maps, and building mapping. Therefore, for each sample residential area there was one dataset regarding spatial elements. In order to solve the problem that one spatial element dataset corresponds to data pertaining to multiple residents’ walking trips in each sampled residential area, the “mode” of multiple residents’ walking trips data in the sample residential area was taken as residents’ walking trips data of the sample residential area so as to achieve a one-to-one correspondence between the residents’ walking trips data and the spatial elements’ data in the research (Figure 2).

### 2.2. Study Characteristics and Study Methods

Residents’ walking trips have spatiotemporal characteristics. The time characteristics are the major evaluation criteria of residential health. The spatial characteristics provide the space for health needs. In this study, residents’ walking trips’ elements were analyzed on the basis of their spatiotemporal characteristics, and the spatial elements of a health-supportive environment in residential streets were analyzed based on residential health needs. To obtain the spatial elements that promote residents’ walking trips, two methods, including correlation analyses and logistic regression, were applied to analyze walking trips elements and spatial elements of health-supportive environments in residential streets (Figure 3).

In 2018, research data were obtained using an on-site survey and a questionnaire survey (Appendix A, Table A1) analyzing residents’ walking trips. The interview and questionnaires were administered to residents who were entering and exiting the residential area at the pedestrian entrances and exits of the access-controlled residential area to ensure that the survey objects were residents of the same residential area. The numbers of questionnaires for each sample residential area were proposed to be 10% of the total numbers of households in the sample residential area, with 2738 questionnaires (Table 1). 2231 valid questionnaires were collected and the statistical data were as shown in Table 2 and Table 3:

People who chose “moderate exercise” were administered a further in-depth interview. A total 375 of people participated in the in-depth interview, 83.1% of the which agreed with and practiced the idea of residents’ walking trips to promote health; they were further investigated regarding their health needs, the spatial environment of residential streets, and the current situation of residents’ walking trips.

### 2.3. Analysis of Elements of Residents’ Walking Trips

Spatiotemporal characteristics are the main research perspective regarding residents’ walking trips. According to the statistics of the questionnaire, the age of the survey objects was mostly concentrated in the middle-aged and young groups (62.1%), of which fixed workers accounted for a relatively high proportion (60.9%).

The main modes of transportation were as follows: walking (35.3%), bicycle (1.4%), electric vehicle (1.6%), motorcycle (4.5%), public transportation (34.3%), private car (22.9%), and walking and public transportation (69.6%).

The time required for getting to work (one way) was within 10 min (8.3%), 10–20 min (14.1%), 20–30 min (27.9%), 30–40 min (23.50%), 40–50 min (13.2%), 50–60 min (9.1%), and more than 60 min (3.9%). Most residents’ one-way commute time to work was within 20–40 min; the smallest group had a one-way commute to work that took more than 60 min. Therefore, taking the daily walking and public transportation commuters as an example, the time–space prism simulation can be more representative for the analysis of elements of residents’ walking trips (The spatiotemporal prism methodis a method of modeling the physiological, physical, and environmental constraints on individual behavior [31]).

Necessary activities refer to activities restricted by activity location and activity time; triggered activities refer to the activities that can be randomly selected within limits. There are many triggered activities that may arise in the process of performing necessary activities. As shown in Figure 4, there are multiple triggered activity locations (such as breakfast shops, stores, supermarkets and other triggered activity locations) in the necessary activities locations (such as schools, homes, workplaces and other necessary activity locations). Residents’ walking trips can be divided into necessary residents’ walking trips with a specific destination and with limited time, and triggered walking trips without a clear destination and with more spatial possibilities.

Residents’ daily walking trips’ time is positively correlated with their health situation [32], which can be used as a scale to evaluate the health-supportive environment in residential streets. Through the questionnaire survey from the ten sampled residential areas, it was found that walking time and the frequency of walking trips were not only limited by the street space, but also affected by residents’ daily schedules. The total individual walking trips’ time within a certain period was determined by the frequency of walking trips and their durations together.

The frequency of walking trips includes the frequency of necessary walking trips and the frequency of triggered walking trips. The duration of walking trips includes the duration of necessary walking trips and the duration of triggered walking trips. Therefore, the elements of residents’ walking trips mainly include four kinds as follows: the frequency of necessary walking trips defined as Y1, the duration of necessary walking trips defined as Y2, the frequency of triggered walking trips defined as Y3, and the duration of triggered walking trips defined as Y4 (Table 4).

### 2.4. Analysis of the Spatial Elements of a Health-Supportive Environment in Residential Streets

#### 2.4.1. Residential Health Needs

Based on residential health needs, this study discusses the elements of residents’ walking trips in residential streets; a health-supportive environment refers to an environment that can meet physical health needs, mental health needs, social adaptability health needs, and moral health needs [33]. Therefore, the residential health needs can be divided into physical health needs, mental health needs, social adaptability health needs, and moral health needs from the perspectives of residents’ walking trips as follows:

Physical health needs mainly refer to a spatial environment that is convenient for walking trips in residential streets; this includes smooth walking paths, good pedestrian access, and bus stations, and an environment that is more convenient for business facilities [32,33].

Mental health needs mainly refer to a spatial environment in residential streets that is conductive to the release of residents’ psychological pressure; a higher density of frontage stores; an appropriate height-to-width ratio of the street; wide pavements; and beautiful street greening [34].

Social adaptability health needs mainly refer to a spatial environment in residential streets that promotes good neighborhood relationships, including higher density of street intersections, multi-function streets, mixed building ages, multi-functional shops near the street, and high walkway flow [35].

Moral health needs mainly refer to a spatial environment in residential streets that protects pedestrians. Pedestrians are a vulnerable group compared to motor vehicles, and there are also relatively vulnerable groups among pedestrians, such as the sick and the disabled. In order to protect pedestrians from the interference of motor vehicles, enough pedestrian crossing facilities, enough outdoor facilities, and sufficient night lighting are needed. In order to protect the walking convenience of the disabled, a perfect blind track system and wheelchair-accessible ramps are needed [36].

#### 2.4.2. The Current Problems in Residential Streets

Based on the on-site interviews and questionnaires, it was found that the spatial environment in the sampled residential streets cannot completely meet the residential walking health needs. From the interviews regarding the two aspects of “the spatial environment promoting residents’ walking trips in residential streets” and “the spatial environment hindering residents’ walking trips in residential streets” (Appendix A, Table A2), four main problems at present were found, as follows:

Path problems of walking trips: low density of the residential street network; unsmooth pedestrian paths; narrow width of pedestrian walkways on streets; and no openings or few openings on the street interface of the building.

Safety problems of walking trips: slippery road surfaces in rainy days; inadequate lighting at night; lack of crossing facilities; illegal occupation of sidewalks.

Environmental problems of walking trips: lack of humanization of hard pavement; poor aesthetic effect of buildings along the street; lack of green landscape; imbalance of height–width ratio of street space; poor environmental sanitation.

Spatial function problems of walking trip streets: lack of shared leisure communication spaces and rest nodes; lack of effective connections between the underlying space of the building and the street; fragmentation of the street space; lack of open shops on the street; lack of places suitable to activities for differently aged.

#### 2.4.3. Spatial Elements of a Health-Supportive Environment in Residential Streets Based on Residential Health Needs

The above four main problems correspond to the health needs of residents’ walking trips, which is an important correspondence for researching the health-supportive environment in residential streets. The spatial elements of a health-supportive environment in residential streets can be divided into convenience, safety, comfort, and diversity (Figure 5).

Convenience: connectivity of residents’ walking trips network, which includes intersection density and public transport network density [37,38,39]; and accessibility of public facilities [37,40], including a comprehensive supermarket coverage radius and a leisure and entertainment coverage radius [41,42,43].

Safety: traffic safety [44,45], which includes the location and length of crosswalks, traffic signs and signals for pedestrians and vehicles, and width of sidewalks [37,46,47]; public security, which includes road monitoring, night lighting, social security, and personal safety; psychological safety [37,48], which includes street closures, facade transparency of ground floor buildings, the proportion of ground floor commercial buildings, and the density of stores along the street [47,49].

Comfort: landscape comfort [33,47], which includes the degree of repair, street greening perfection, street scale, and landscape quality [50,51,52,53,54]; travel comfort, which includes the perfection of supporting facilities, network connectivity, and near-line building rate [55,56].

Diversity: the distance between leisure and entertainment shopping destinations [41,42] which includes the density of commercial facilities and the density of leisure and entertainment facilities [51,57,58]; various land-use models, which includes the mixed use of frontage stores and the varieties of street forms [45,59,60].

#### 2.4.4. The Spatial Element Data Collection of a Health-Supportive Environment in Residential Streets

On the basis of research relevant to this study, the spatial element data were collected through on-site interviews, street view maps, and building surveying and mapping. The specific process was as follows:

First of all, the relevant data of spatial elements were obtained from the CAD file of the current land use situation of Nan’an District, Chongqing City, and coordinated with streetscape maps, Baidu maps and Tencent maps to obtain relevant data of spatial elements (Appendix A, Table A3).

Secondly, through on-site interviews, the obtained data were supplemented, perfected, and amended to improve the accuracy and the reliability of the original data.

Finally, the spatial elements of the health-supportive environment in residential streets were subdivided, numbered, and assigned in the study (Appendix A, Table A4).

### 2.5. The Quantitative Analysis Method of Spatial Elements for a Health-Supportive Environment that Promotes Walking Trips in Residential Streets

The spatial elements of a health-supportive environment are defined as X variables (Appendix A, Table A4), and the elements of residents’ walking trips are defined as Y variables (Appendix A, Table A4). Data processing and data modeling were carried out for X and Y variables. The quantitative analysis of spatial elements was divided into two steps as follows (Figure 6).

Step 1: Correlation analysis.

To explore the spatial elements of a health-supportive environment in residential streets that promotes residents’ walking trips, we had to find a X variable that had a significant correlation with the Y variable, using the methods of a contingency coefficient analysis, a Spearman’s coefficient analysis, and a second choice of variable clustering. Different variable types corresponded to different correlation analysis methods and judgment standards (Table 5).

Step 2: Ordered logistic regression analysis.

For the X and Y variables filtered in step 1, the X variable is defined as an independent variable and the Y variable is defined as a dependent variable. Quantitative analysis and model fitting equation calculations can be carried out using ordered logistic regression analysis. The model and specific hypothesis of the ordered logistic regression are cumulative ordered logistic models, which need to be calculated using a cumulative probability formula (Formulas (1)–(4)). For some X variables that cannot be calculated by the formula calculation, an OR value can be used to judge their influence on the Y variables (Formula (5)).

In the cumulative probability formula, the dependent variable *j* was set as the ordered variable of K pieces of grades, and K pieces of grades was represented by 1, 2 … K. Grade K is divided into two categories: {1, J} and {*j* + 1, K}, while *j* is the cutting point; the logit is defined on the basis of these two types: the logarithm of the advantages of the accumulated probability of the last K–J level and the accumulated probability of the first *j* level; the model is thus called the accumulated advantage model, in which the model assumes that the level of the dependent variable is the same regardless of classification. That is, the regression coefficient of the independent variable is independent of the partition point J, which is the cumulative probability ratio of the former to the latter:(1)P(y ≤ j|x)=P(y=1|x)+…+P(y=j|x)
(2)$$\mathrm{Logit}\;P_{j}=\mathrm{Logit}\left[P\left(y>j|x\right)\right]=\ln\frac{P\left(y>j|x\right)}{1-P\left(y>j|x\right)}$$
(3)$$\mathrm{Logit}\;P_{j}=\mathrm{Logit}\left[P\left(y>j|x\right)\right]=-\alpha_{j}+\overset{p}{\underset{i=1}{\sum }}\beta_{i}x_{i}$$
(4)$$P\left(y\;\leq \;j|x\right)=\frac{1}{1+\exp\left(-\alpha_{j}+\sum_{i=1}^{p}\beta_{i}x_{i}\right)}$$

The OR value indicates the ratio of the *y* value to the ratio of one or more grades (the logarithm of the dominance ratio) for each unit of change of independent variable X*i*; that is, how many times does the *y* value increase for each unit increase of the independent variable X*i*.): (5)$$\mathrm{OR}=\exp\left(\beta_{i}\right)$$

## 3. Results

In the process of the statistical data analysis, the data were analyzed from the correlation analysis and the ordered logistic regression in this study. The research results include the results of the correlation analysis and the results of the ordered logistic regression analysis.

### 3.1. Correlation Analysis Results

The significance level test was set to 0.05, and the correlation analysis conclusions were as follows:

The spatial elements needed to improve the frequency of necessary residents’ walking trips (Y1) were as follows (Table 6): density of pedestrian access (XB1), convenience of service stores (XB4), density of bus routes (XB11), and average pedestrian access distance (XC3).

The spatial elements needed to increase the duration of necessary residents’ walking trips (Y2) were as follows (Table 6): mixed uses of service store kinds (XA3), density of street intersections (XA15), density of street directions (XA16), mixed proportion of differently aged of residential buildings (XA17), ratios of old residential buildings (XA19), and average pedestrian access distance (XC3).

The spatial elements needed to improve the frequency of triggered residents’ walking trips (Y3) were as follows (Table 6): mixed uses of leisure and entertainment store kinds (XA4), population density (XA20), density of pedestrian access (XB1), convenience of leisure and entertainment store kinds (XB10), average pedestrian access distance (XC3), square area within a 500 m walking distance (XD1), and distance to the nearest garden (XD2).

The spatial elements needed to increase the duration of triggered residents’ walking trips (Y4) were as follows (Table 6): mixed uses of service store kinds (XA3), distribution density of leisure and entertainment stores (XA10), density of street intersection (XA15), density of street directions (XA16), ratios of old residential buildings (XA19), density of pedestrian access (XB1), convenience of service stores (XB4), convenience of service store kinds (XB9), near-line rate of roadside buildings (XC2), average pedestrian access distance (XC3), and green shade ratio (XD6).

In summary, the conclusions are as follows: safety is the basis of walking trips; this has an impact on the frequency of necessary walking trips, the duration of necessary walking trips, the frequency of triggered walking trips and the duration of triggered walking trips. Meanwhile, convenience has the greatest influence on the frequency of necessary and triggered walking trips. Diversity has the greatest influence on the duration of necessary and triggered walking trips. Comfort has the greatest influence on the frequency and duration of triggered walking trips.

### 3.2. Ordered Logistic Regression Analysis Results

The significance level test was set to 0.05, the Y variable was defined as the dependent variable and the X variable was defined as the independent variable. The ordered logistic regression analysis conclusions are as follows:

The spatial elements needed to improve the frequency of necessary residents’ walking trips (Y1) are as follows (Table 7): density of pedestrian access (XB1), density of bus routes (XB11), and average pedestrian access distance (XC3).

The spatial elements needed to increase the duration of necessary residents’ walking trips (Y2) are as follows (Table 7): density of street intersection (XA15), and mixed proportion of differently aged residential buildings (XA17).

The spatial elements needed to improve the frequency of triggered residents’ walking trips (Y3) are as follows (Table 7): square area within a 500 m walking distance (XD1) and distance to the nearest garden (XD2).

The spatial elements needed to increase the duration of triggered residents’ walking trips (Y4) are as follows (Table 7): near-line rate of roadside buildings (XC2), average pedestrian access distance (XC3), green shade ratio (XD6).

Meanwhile, according to the β value in Table 7, the models of Y1, Y2, Y3, and Y4 have been calculated using Formulas (1)–(4) (Section 2.5) of the cumulative probability formula.

#### 3.2.1. Ordered Logistic Regression Model of the Frequency of Necessary Residents’ Walking Trips (Y1)

According to the values in Table 4 (Section 2.3), one to two times per week are value “1”, three to six times per week are value “2”, one time per day is value “3”, and two times per day and above are value “4”; hence the frequency of necessary walking trips (Y1) was divided into four levels according to the β value in Table 7. The model cutting points (J) are three and the supposed models when J is 1, 2, or 3 are as follows:

Cutting model 1: {“1”}, {“2”, “3”, “4”}

Cutting model 2: {“1”, “2”}, {“3”, “4”}

Cutting model 3: {“1”, “2”, “3”}, {“4”}

The cumulative probability of Y1 (Formulas (6)–(10)) can be calculated using Formulas (1)–(4) (Section 2.5) of the cumulative probability formula.
(6)$$\mathrm{Logit}\;P_{j}=-\alpha_{j}+1.104\mathrm{XB}1+0.385\mathrm{XB}11-0.018\mathrm{XC}3$$
(7)$$P\left(Y1=\;“1”\right)=\;P\left(Y1\;\leq \;”1”\right)=\frac{1}{1+e^{4.094+1.104\mathrm{XB}1+0.385\mathrm{XB}11-0.018\mathrm{XC}3}}$$
(8)$$\begin{array}{cc} P\left(Y1=\;“2”\right)=\; & P\left(Y1\;\leq \;“2”\right)-\;P\left(Y1\;\leq “1”\right)\\ & \;=\frac{1}{1+e^{0.727+1.104\mathrm{XB}1+0.385\mathrm{XB}11-0.018\mathrm{XC}3}}\\ & \;-\frac{1}{1+e^{4.094+1.104\mathrm{XB}1+0.385\mathrm{XB}11-0.018\mathrm{XC}3}} \end{array}$$
(9)$$\begin{array}{cc} P\left(Y1=\;“3”\right)=\; & P\left(Y1\;\leq \;“3”\right)-\;P\left(Y1\;\leq \;“2”\right)\\ & \;=\frac{1}{1+e^{-0.824+1.104\mathrm{XB}1+0.385\mathrm{XB}11-0.018\mathrm{XC}3}}\\ & \;-\frac{1}{1+e^{0.727+1.104\mathrm{XB}1+0.385\mathrm{XB}11-0.018\mathrm{XC}3}} \end{array}$$
(10)$$P\left(Y1=\;“4”\right)=1-\;P\left(Y1\;\leq \;“3”\right)=1-\frac{1}{1+e^{-0.824+1.104\mathrm{XB}1+0.385\mathrm{XB}11-0.018\mathrm{XC}3}}$$

#### 3.2.2. Ordered Logistic Regression Model of the Duration of Necessary Residents’ Walking Trips (Y2)

According to the values in Table 4 (Section 2.3), 0 to 5 min are value “1”, 5 to 15 min are value “2”, 15 to 30 min are value “3”, 30 to 45 min are value “4”, 45 to 60 min are value “5”, and 60 min and above value are “6”. The duration of necessary walking trips (Y2) has six levels in theory; however, the value “1” does not exist in the sampled residential area; thus according to the β value in Table 7, there are five levels in the sample data and there are four model cutting points (J). The models can be supposed when J is 1, 2, 3, or 4 as follows:

Cutting model 1: {“2”}, {“3”, “4”, “5”, “6”};

Cutting model 2: {“2”, “3”}, {“4”, “5”, “6”};

Cutting model 3: {“2”, “3”, “4”}, {“5”, “6”};

Cutting model 4: {“2”, “3”, “4”, “5”}, {“6”}.

The cumulative probability of Y2 (Formulas (11)–(16)) can be calculated by the cumulative probability formula of Formulas (1)–(4) (Section 2.5) as follows:(11)$$\mathrm{Logit}\;P_{j}=-\alpha_{j}+1.254\mathrm{XA}15+7.804\mathrm{XA}17$$
(12)$$P\left(Y2=\;“2”\right)=\;P\left(Y2\;\leq \;“2”\right)=\frac{1}{1+e^{-1.912+1.254\mathrm{XA}15+7.804\mathrm{XA}17}}$$
(13)$$\begin{array}{cc} P\left(Y2=\;“3”\right)=\; & P\left(Y2\;\leq \;“3”\right)-\;P\left(Y2\;\leq \;“2”\right)\\ & \;=\frac{1}{1+e^{-2.952+1.254\mathrm{XA}15+7.804\mathrm{XA}17}}-\frac{1}{1+e^{-1.912+1.254\mathrm{XA}15+7.804\mathrm{XA}17}} \end{array}$$
(14)$$\begin{array}{cc} P\left(Y2=\;“4”\right)=\; & P\left(Y2\;\leq \;“4”\right)-\;P\left(Y2\;\leq \;“3”\right)\\ & =\frac{1}{1+e^{-5.484+1.254\mathrm{XA}15+7.804\mathrm{XA}17}}-\frac{1}{1+e^{-2.952+1.254\mathrm{XA}15+7.804\mathrm{XA}17}} \end{array}$$
(15)$$\begin{array}{cc} P\left(Y2=\;“5”\right)=\; & P\left(Y2\;\leq \;“5”\right)-\;P\left(Y2\;\leq \;“4”\right)\\ & \;=\frac{1}{1+e^{-8.541+1.254\mathrm{XA}15+7.804\mathrm{XA}17}}-\frac{1}{1+e^{-5.484+1.254\mathrm{XA}15+7.804\mathrm{XA}17}} \end{array}$$
(16)$$P\left(Y2=\;“6”\right)=1-\;P\left(Y2\;\leq \;“5”\right)=1-\frac{1}{1+e^{-8.541+1.254\mathrm{XA}15+7.804\mathrm{XA}17}}$$

#### 3.2.3. Ordered Logistic Regression Model of the Frequency of Triggered Residents’ Walking Trips (Y3)

According to the values in Table 4 (Section 2.3), one to two times per week are value “1”, three to six times per week ate value “2”, one time per day is value “3”, and two times per day and above are value “4”; the frequency of triggered walking trips (Y3) was divided into four levels according to the to β value in Table 7. There are three model cutting points (J) and the models supposed when J is 1, 2, or 3 are as follows:

Cutting model 1: {“1”}, {“2”, “3”, “4”}

Cutting model 2: {“1”, “2”}, {“3”, “4”}

Cutting model 3: {“1”, “2”, “3”}, {“4”}

The cumulative probability of Y3 (Formulas (17)–(21)) can be calculated using the cumulative probability formula of Formulas (1)–(4) (Section 2.5) as follows:(17)$$\mathrm{Logit}\;P_{j}=-\alpha_{j}+1.339\mathrm{XD}1-0.004\mathrm{XD}2$$
(18)$$P\left(Y3=\;“1”\right)=\;P\left(Y3\;\leq \;“1”\right)=\frac{1}{1+e^{5.214+1.339\mathrm{XD}1-0.004\mathrm{XD}2}}$$
(19)$$\begin{array}{cc} P\left(Y3=\;“2”\right)=\; & P\left(Y3\;\leq \;“2”\right)-\;P\left(Y3\;\leq \;“1”\right)\\ & \;=\frac{1}{1+e^{2.462+1.339\mathrm{XD}1-0.004\mathrm{XD}2}}-\frac{1}{1+e^{5.214+1.339\mathrm{XD}1-0.004\mathrm{XD}2}} \end{array}$$
(20)$$\begin{array}{cc} P\left(Y3=\;“3”\right)=\; & P\left(Y3\;\leq \;“3”\right)-\;P\left(Y3\;\leq \;“2”\right)\\ & \;=\frac{1}{1+e^{0.254+1.339\mathrm{XD}1-0.004\mathrm{XD}2}}-\frac{1}{1+e^{2.462+1.339\mathrm{XD}1-0.004\mathrm{XD}2}} \end{array}$$
(21)$$P\left(Y3=\;“4”\right)=1-\;P\left(Y3\;\leq \;“3”\right)=1-\frac{1}{1+e^{0.254+1.339\mathrm{XD}1-0.004\mathrm{XD}2}}$$

#### 3.2.4. Ordered Logistic Regression Model of the Duration of Triggered Residents’ Walking Trips (Y4)

According to the values in Table 4 (Section 2.3), 0 to 5 min are value “1”, 5 to 15 min are value “2”, 15 to 30 min are value “3”, 30 to 45 min are value “4”, 45 to 60 min are value “5”, and 60 min and above are value “6”. The duration of triggered walking trips (Y4) has six levels in theory; however, the value “6” did not exist in the sampled residential area. According to the β value in Table 7, there are five levels in the sample data and there are four model cutting points (J). The models can be supposed when J is 1, 2, 3, or 4 as follows:

Cutting model 1: {“1”}, {“2”, “3”, “4”, “5”}

Cutting model 2: {“1”, “2”}, {“3”, “4”, “5”}

Cutting model 3: {“1”, “2”, “3”}, {“4”, “5”}

Cutting model 4: {“1”, “2”, “3”, “4”}, {“5”}

The cumulative probability of Y4 (Formulas (22)–(27)) can be calculated using Formulas (1)–(4) (Section 2.5) of the cumulative probability formula, as follows:(22)$$\mathrm{Logit}\;P_{j}=-\alpha_{j}+6.320\mathrm{XC}2-0.018\mathrm{XC}3+3.506\mathrm{XD}6$$
(23)$$P\left(Y4=\;“1”\right)=\;P\left(Y4\;\leq \;“1”\right)=\frac{1}{1+e^{4.148+6.320\mathrm{XC}2-0.018\mathrm{XC}3+3.506\mathrm{XD}6}}$$
(24)$$\begin{array}{cc} P\left(Y4=\;“2”\right)=\; & P\left(Y4\;\leq \;“2”\right)-\;P\left(Y4\;\leq \;“1”\right)\\ & \;\;=\frac{1}{1+e^{1.861+6.320\mathrm{XC}2-0.018\mathrm{XC}3+3.506\mathrm{XD}6}}-\frac{1}{1+e^{4.148+6.320\mathrm{XC}2-0.018\mathrm{XC}3+3.506\mathrm{XD}6}} \end{array}$$
(25)$$\begin{array}{cc} P\left(Y4=\;“3”\right)=\; & P\left(Y4\;\leq \;“3”\right)-\;P\left(Y4\;\leq \;“2”\right)\\ & \;\;=\frac{1}{1+e^{-0.474+6.320\mathrm{XC}2-0.018\mathrm{XC}3+3.506\mathrm{XD}6}}-\frac{1}{1+e^{1.861+6.320\mathrm{XC}2-0.018\mathrm{XC}3+3.506\mathrm{XD}6}} \end{array}$$
(26)$$\begin{array}{cc} P\left(Y4=\;“4”\right)=\; & P\left(Y4\;\leq \;“4”\right)-\;P\left(Y4\;\leq \;“3”\right)\\ & \;\;=\frac{1}{1+e^{-3.064+6.320\mathrm{XC}2-0.018\mathrm{XC}3+3.506\mathrm{XD}6}}-\frac{1}{1+e^{-0.474+6.320\mathrm{XC}2-0.018\mathrm{XC}3+3.506\mathrm{XD}6}} \end{array}$$
(27)$$P\left(Y4=\;“5”\right)=1-\;P\left(Y4\;\leq \;“4”\right)=1-\frac{1}{1+e^{-3.064+6.320\mathrm{XC}2-0.018\mathrm{XC}3+3.506\mathrm{XD}6}}$$

## 4. Discussion

### 4.1. Insights from the Correlation Analysis regarding the Construction of a Health-Supportive Environment in Residential Streets

There are many relatively comprehensive studies on the spatial elements of residential streets that promote residents’ walking trips: the European alpha environmental questionnaire from 2004 and the US residents’ walking trips index from 2007 and 2013 [30]. However, the impact of spatial elements on residents’ walking trips has not yet been fully investigated. Based on the results of the correlation analysis, this study effectively quantifies the impact degree of each spatial element on walking trips and analyzes the impact degree by the frequency level (Figure 7).

There are some insights from the correlation analysis regarding health-supportive environmental construction in residential streets:

First, pedestrian access has a great impact on the residents’ walking trips in residential streets. Non-gated residential communities and small-scale residential communities can effectively promote walking trips. Therefore, more pedestrian accesses will be considered in urban planning and construction.

Second, roadside buildings, street intersections, service facilities, and so on have great influence on walking trips. The density of roadside buildings should be increased, T-shaped road breaking roads should be avoided, the openness of roadside buildings should be improved, buildings with historical characteristics should be maintained, the functions of roadside buildings should be enriched, and the distribution density and the distribution species of service facilities should be increased in the process of urban planning and construction.

Thirdly, public open space and the types of leisure and entertainment also have a certain impact on walking trips. Increasing the distribution density and distribution species of leisure and entertainments facilities should be considered, as well as improving the greening coverage and increasing the density of parks and the squares during urban planning and construction.

Finally, the density of bus routes and population density also have a certain impact on walking trips. They are part of the contents of urban master planning. The density of bus lines should be increased and the population density should be improved in urban master planning in the future.

### 4.2. Application of the Ordered Logistic Regression Analysis to a Health-Supportive Environment in Residential Streets

First, a health-supportive environment evaluation standard on the basis of walking trip time as a health standard was built; second, ten sample residential areas were evaluated by using walking trips time as a health standard; and finally, the effective adjustment of the spatial elements used the ordered logistic regression formula.

#### 4.2.1. Establishing a Health Evaluation Standard

Residents’ physical health was measured by the number of walking steps or daily walking time [61,62]; the walking health standard was set in this study as 60 min or more of walking every day [63,64,65,66].

#### 4.2.2. Health Evaluation of the Ten Sampled Residential Areas

Seven sampled residential areas were unqualified during evaluating in ten sampled residential areas (Table 8).

#### 4.2.3. The Effective Adjustment of the Spatial Elements Used in the Ordered Logistic Regression Formula

The spatial elements of seven unqualified sample residential areas were adjusted effectively by using the ordered logistic regression formula to increase the residents’ walking trip times per day.

Jinyang Roman holiday residential street was taken as an example:

Model test (Table 9):

The predicted data of the model were consistent with the real data, and the model can be further analyzed.

Adjustment of residents’ walking trips level (Table 10).

When the frequency level of residents’ necessary walking trips (Y1) and the frequency level of residents’ triggered walking trips (Y3) were adjusted effectively, residents’ walking trip times per day were adjusted from 17 to 60 min.

Spatial element index adjustment (Table 11):

The density of pedestrian access, the density of bus routes, the average pedestrian access distance, the distance to the nearest block parks and the area of squares accessibility within a 500 m distance were adjusted with the ordered logistic regression Formulas (7)–(10) (Section 3.2.1) and Formulas (18)–(21) (Section 3.2.3). However, this adjustment is only for theoretical guidance and is combined with the current situation in the practice application.

### 4.3. Research Limitations

There are certain errors in the statistical data of residents’ walking trips, which were gathered by means of on-site interviews and questionnaires. Therefore, for further research, large data and mobile devices are considered for data sample collection of residents’ walking trips.

The residents who enter and exit the residential area at the pedestrian access points are mobile; this survey did not conduct more in-depth household surveys. Therefore, it is true that more than one person in the same household may have participated in the research, but it is the small probability event.

There is no direct related data analysis between the spatial elements of a health-supportive environment and residents’ health status.

In the process of the quantification of abstract data and the statistical operation of abstract data, it is inevitable to simplify and neglect the personality of residents, such as residents’ age and gender, which led to some deviations in the results.

The reconstruction plan proposed by the regression model is only for theoretical guidance, and there may be some obstacles in practice, such as the increase of pedestrian access and bus routes, which need the coordination of the government, community, and residents.

### 4.4. Future Research

#### 4.4.1. Specific Research for Particular Groups

The walking trip characteristics of residents of different genders or different age groups are different. The relationship between residents’ walking trips and residential street space environment can be studied according to different genders or different age groups in the future.

#### 4.4.2. Research in Depth on a Sample Residential Area

The transformation strategy guided by theory is applied to the practice of residential street environment construction. Through a more in-depth household survey and the feedback of the project, the optimization and modification of the health-supportive environment for the health of the residential streets to promote walking trips are continuously strengthened.

#### 4.4.3. Follow-Up Sample Population in the Long Term

After using the ordered logistic regression model to adjust the spatial elements of the sample residential area, the impact of the adjusted spatial elements on the walking trip times and the health of sample residential area residents can be explored.

## 5. Conclusions

The spatial elements of a health-supportive environment in residential streets that promote residents’ walking trips and the transformation strategy with the support of a data model for the improvement of residents’ walking trips’ environment in residential streets were studied in this paper mainly from three angles, as follows:

Based on the spatiotemporal characteristics of residents’ walking trips, the residents’ walking trips’ factors were divided into necessary walking trips’ frequency, necessary walking trips’ time, triggering walking trips’ frequency, and triggering walking trips’ time.

Based on the residents’ health needs, the spatial elements of the health-supportive environment in residents’ streets were analyzed and the spatial elements were refined from four angles: convenience, safety, comfort, and diversity.

Based on a correlation analysis and an ordered logistic regression analysis, this paper studied the spatial elements of a health-supportive environment for promoting residents’ walking trips in residential streets, established an ordered logistic regression model, and improved residents’ daily walking trips’ time by adjusting spatial elements in reverse.

The study found that the main actions needed are as follows: improving the density of pedestrian access and the density of bus routes in residential areas to improve the convenience of residential streets; improving the near-line rate of roadside buildings and the average pedestrian access distance to increase the safety of residential streets; increasing the square area within a 500 m walking distance, the distance to the nearest garden, and the green shade ratio to improve the comfort of residential streets; increasing the density of street intersections and the mixed proportion of differently aged residential buildings to improve the diversity of residential streets.

## Figures and Tables

**Figure 1 ijerph-17-05198-f001:**
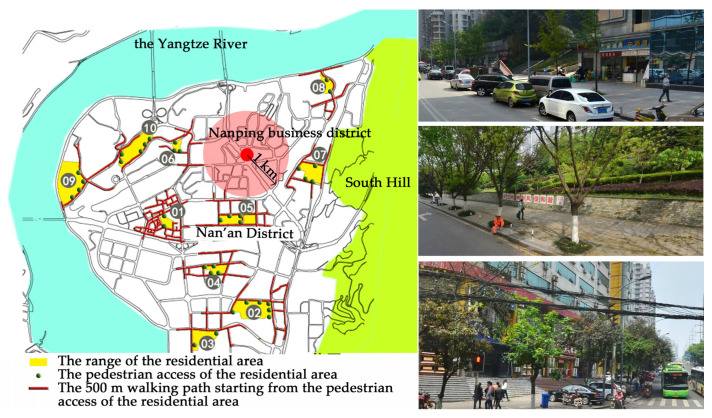
Sample residential areas.

**Figure 2 ijerph-17-05198-f002:**
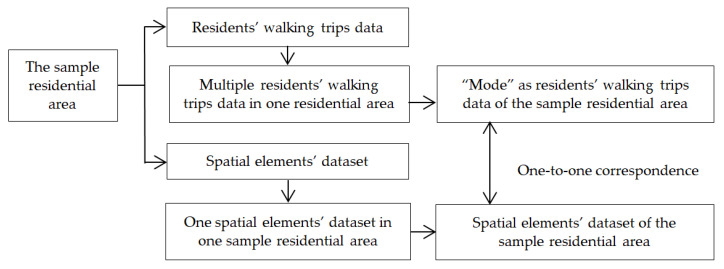
One-to-one data correspondence.

**Figure 3 ijerph-17-05198-f003:**
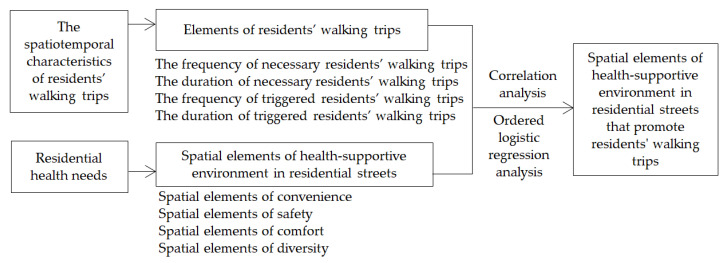
Study characteristics.

**Figure 4 ijerph-17-05198-f004:**
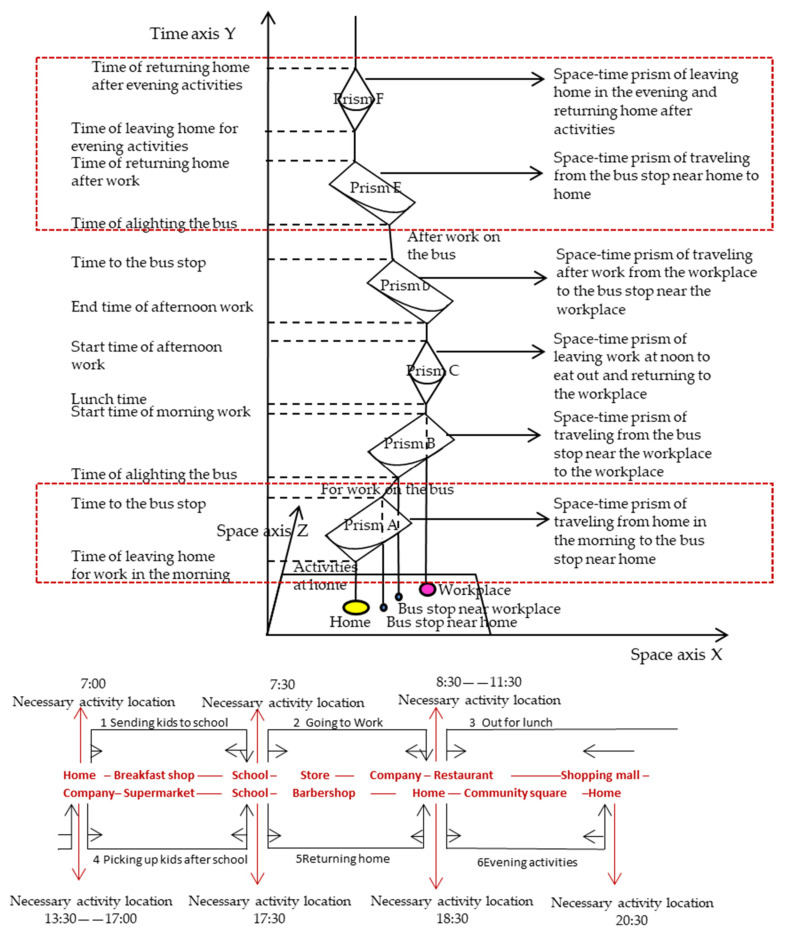
Space-time prism of commuting by bus on workdays.

**Figure 5 ijerph-17-05198-f005:**
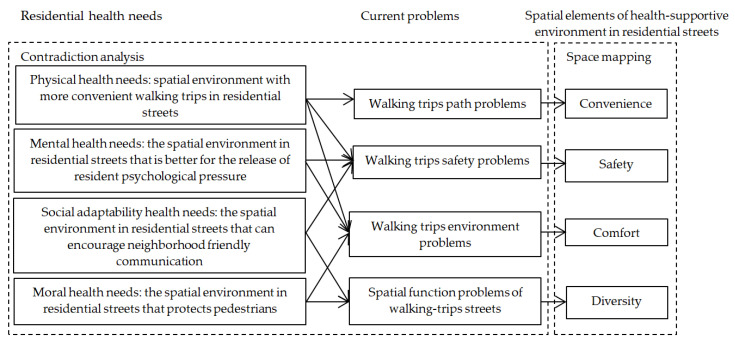
Formative logic of spatial elements of a health-supportive environment in residential streets.

**Figure 6 ijerph-17-05198-f006:**
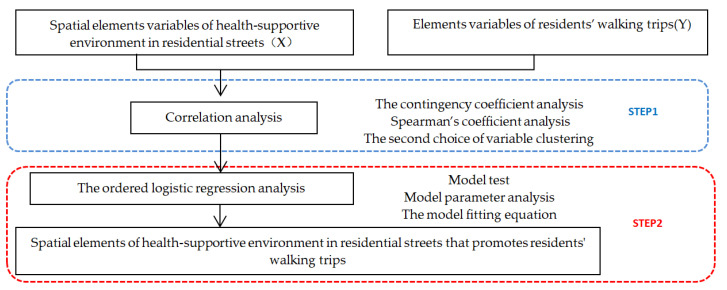
Steps of the quantitative analysis.

**Figure 7 ijerph-17-05198-f007:**
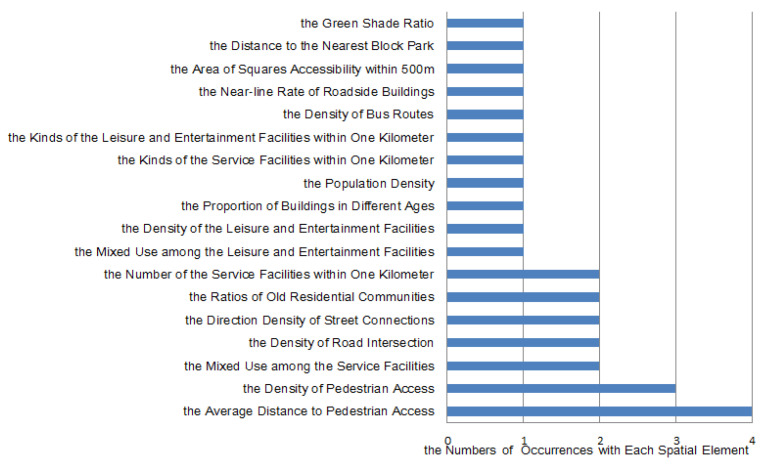
The impact of each spatial element on walking trips.

**Table 1 ijerph-17-05198-t001:** Basic information of the sample residential areas ^1^.

No.	Name of the Sample Residential Area	Total Number of Households	Total Number of Residents	Total Number of Questionnaires
01	Yifeng Garden	810	2384	81
02	Huilongwan Community	3647	11,678	365
03	Yajule International Garden	3275	10,472	328
04	Jinyang Roman Holiday	2700	8541	270
05	Wanshou Garden	720	2307	72
06	Changjiang Village Resettlement House	610	1954	61
07	Kangde Capitol Hill	6000	19,230	600
08	Zone A of Haitang xiao yue	3335	10,643	334
09	Phase I of Rongqiao City	3066	9611	307
10	Xiangmi Mountain on Rongqiao Peninsula	3200	10,245	320
Total		27,363	87,065	2738

^1^ Note: residents are mobile, the data above only represent the data during the study.

**Table 2 ijerph-17-05198-t002:** Basic statistics of questionnaire respondents.

Categories and Statistical Results	A	B	C	D	E
Gender (%)	Female	Male	-	-	-
	57%	43%	-	-	-
Age (%)	7–17 years	18–40 years	41–66 years	Over 66 years	-
	13.7%	40.3%	34.7%	10.1%	-
Education background (%)	Junior high school	Senior high school	Junior college	University undergraduate	Postgraduate and above
	16.6%	37.5%	22.2%	20.9%	2.8%
Occupation (%)	Students	Full-time work	Part-time work	No occupation	Retirees
	7.1%	60.9%	5.9%	11.1%	15%
Positions	Ordinary Staff	Middle-level managers	Senior managers	Self-employed persons	Others
	61.7%	20.6%	1.6%	4.4%	11.7%

**Table 3 ijerph-17-05198-t003:** Methods of controlling body weight, blood sugar, blood pressure and blood lipids.

Categories and Statistical Results	A	B	C	D	E	F
Weight control (%)	Control diet	Low fat diet	Low-calorie diet	Moderate exercise	Drugs	Others
	25.4%	18.1%	11%	35.7%	5.4%	4.4%
Blood pressure control (%)	Regular medication	Irregular medication	Diet control	Moderate exercise	Others	-
	10.2%	23.2%	18.2%	45.3%	3.1%	-
Blood glucose control (%)	Regular medication	Irregular medication	Diet control	Moderate exercise	Others	-
	8.5%	30.4%	16.4%	43.2%	1.5%	-
Blood lipids control (%)	Regular medication	Irregular medication	Diet control	Moderate exercise	Others	-
	14.3%	23.4%	18.4%	43.2%	1.5%	-

**Table 4 ijerph-17-05198-t004:** Classification of different elements on residents’ walking trips.

Activity Type	Characteristics	Elements of the Classification of Residents’ Walking Trips	Description of Value Assignment ^1^	Variable Type
Necessary activities	Activities restricted by activity location and activity time	Frequency of residents’ necessary walking trips (Y1)	One to two times per week—value “1” Three to six times per week—value “2” One time per day—value “3” Two times per day and above—value “4”	Ordinal variable
		Duration of residents’ necessary walking trips (Y2)	0 to 5 min—value “1”, 5 to 15 min—value “2”, 15 to 30 min—value “3”, 30 to 45 min—value “4”, 45 to 60 min—value “5” 60 min and above—value “6”	Ordinal variable
Triggered activities	Activities that can be randomly selected within limits	Frequency of residents’ triggered walking trips (Y3)	One to two times per week—value “1” Three to six times per week—value “2” One time per day—value “3” Two times per day and above—value “4”	Ordinal variable
		Duration of residents’ triggered walking trips (Y4)	0 to 5 min—value “1”, 5 to 15 min—value “2”, 15 to 30 min—value “3”, 30 to 45 min—value “4”, 45 to 60 min—value “5” 60 min and above—value “6”	Ordinal variable

^1^ The walking frequency and walking time are assigned according to the ordinal variables.

**Table 5 ijerph-17-05198-t005:** Methods of correlation analysis for different variable types ^1^.

Variable Type	Continuous Variable	Ordinal Variable	Categorical Variable
Continuous variable	Pearson coefficient	Spearman’s coefficient	Eta coefficient or difference analysis
Ordinal variable	Spearman’s coefficient	Spearman’s coefficient	Contingency coefficient, Phi (Φ) coefficient, or difference analysis
Categorical variable	Eta coefficient or difference analysis	Contingency coefficient, Phi (Φ) coefficient, or difference analysis	Contingency coefficient or Phi (Φ) coefficient

^1^ Continuous variable: the data are made up of discrete numbers, such as the height variable (160 cm, 163 cm, 180 cm, 176 cm); ordinal variable: the data are a group of numbers, and the grades between the numbers increase or decrease, such as the satisfaction variable (“1” dissatisfaction, “2” neutral, “3” satisfaction); category variable: the data are a group of numbers, and these numbers are not related, such as the gender variable (“1” male, “2” female).

**Table 6 ijerph-17-05198-t006:** Methods of correlation analysis for different variable types ^1^.

Spatial Elements of Health-Supportive Environment in Residential Streets That Promote Residents’ Walking Trips (X)			Significance Level	Correlation Coefficient	Category
Spatial elements of health-supportive environment in residential streets (X) that promote the frequency of necessary residents’ walking trips (Y1)	XB1	Density of pedestrian access	0.018	0.725 *	Convenience
	XB4	Convenience of service stores	0.049	0.632 *	
	XB11	Density of bus routes	0.022	0.707 *	
	XC3	Average pedestrian access distance	0.024	−0.700 *	Safety
Spatial elements of health-supportive environment in residential streets (X) that promote the duration of residents’ necessary walking trips (Y2)	XA3	Mixed uses of service store kinds	0.004	0.820 **	Diversity
	XA15	Density of street intersections	0.030	0.681 *	
	XA16	Density of street directions	0.024	0.700 *	
	XA17	Mixed proportion of differently aged residential buildings	0.010	0.765 **	
	XA19	Ratios of old residential buildings	0.010	0.768 **	
	XC3	Average pedestrian access distance	0.014	−0.743 *	Safety
Spatial elements of health-supportive environment in residential streets (X) that promote the frequency of residents’ triggered walking trips (Y3)	XA4	Mixed uses of leisure and entertainment store kinds	0.004	0.814 **	Diversity
	XA20	Population density	0.021	0.711 *	
	XB1	Density of pedestrian access	0.048	0.636 *	Convenience
	XB10	Convenience of leisure and entertainment store kinds	0.042	0.650 *	
	XC3	Average pedestrian access distance	0.033	−0.673 *	Safety
	XD1	Square area within a 500 m walking distance	0.042	0.650 *	Comfort
	XD2	Distance to the nearest garden	0.037	−0.661 *	
Spatial elements of health-supportive environment in residential streets (X) that promote the duration of residents’ triggered walking trips (Y4)	XA3	Mixed uses of service store kinds	0.000	0.904 **	Diversity
	XA10	Distribution density of leisure and entertainment stores	0.044	0.644 *	
	XA15	Density of street intersections	0.000	0.898 **	
	XA16	Density of street directions	0.001	0.879 **	
	XA19	Ratio of old residential buildings	0.020	0.715 *	
	XB1	Density of pedestrian access	0.018	0.725 *	Convenience
	XB4	Convenience of service stores	0.005	0.805 **	
	XB9	Convenience of service store kinds	0.003	0.832 **	
	XC2	Near-line rate of roadside buildings	0.036	0.665 *	Safety
	XC3	Average pedestrian access distance	0.003	−0.836 **	Safety
	XD6	Green shade ratio	0.043	0.678 *	Comfort

^1^ *: 0.01 < *p*-values < 0.05, a certain correlation between the two variables. **: *p*-values ≤ 0.01, a high correlation between the two variables.

**Table 7 ijerph-17-05198-t007:** Ordered logistic regression analysis ^1^.

Spatial Elements of Health-Supportive Environment in Residential Streets That Promote Residents’ Walking Trips (X)			β	Significance Level	“OR” Value	Category
Spatial elements of health-supportive environment in residential streets (X) that promote the frequency of necessary residents’ walking trips (Y1)	Constant term	α1 (Y1 = “1”)	−4.094	0.001	-	-
		α2 (Y1 = “2”)	−0.727	0.005	-	-
		α3 (Y1 = “3”)	0.824	0.001	-	-
	XB1 (Density of pedestrian access)		1.104	0.044	3.016	Convenience
	XB11 (Density of bus routes)		0.385	0.021	1.470	Convenience
	XC3 (Average pedestrian access distance)		−0.018	0.007	0.982	Safety
Spatial elements of health-supportive environment in residential streets (X) that promote the duration of residents’ necessary walking trips (Y2)	Constant term	α2 (Y2 = “2”)	1.912	0.080	-	-
		α3 (Y2 = “3”)	2.952	0.032	-	-
		α4 (Y2 = “4”)	5.484	0.034	-	-
		α5 (Y2 = “5”)	8.541	0.008	-	-
	XA15 (Density of street intersection)		1.254	0.046	3.504	Diversity
	XA17 (The mixed proportion of differently aged residential buildings)		7.804	0.049	2.450	Diversity
Spatial elements of health-supportive environment in residential streets (X) that promote the frequency of residents’ triggered walking trips (Y3)	Constant term	α1 (Y3 = “1”)	−5.214	0.010	-	-
		α2 (Y3 = “2”)	−2.462	0.041	-	-
		α3 (Y3 = “3”)	−0.254	0.292	-	-
	XD1 (Square area within a 500 m walking distance)		1.339	0.036	3.815	Comfort
	XD2 (Distance to the nearest garden)		−0.004	0.002	0.996	Comfort
Spatial elements of health-supportive environment in residential streets (X) that promote the duration of residents’ triggered walking trips (Y4)	Constant term	α2 (Y4 = “1”)	−4.148	0.397	-	-
		α3 (Y4 = “2”)	−1.861	0.675	-	-
		α4 (Y4 = “3”)	0.474	0.918	-	-
		α5 (Y4 = “4”)	3.064	0.512	-	-
	XC2 (Near-line rate of roadside buildings)		6.320	0.042	555.573	Safety
	XC3 (Average pedestrian access distance)		−0.018	0.042	0.982	Safety
	XD6 (Green shade ratio)		3.506	0.038	33.315	Comfort

^1^ the values of “1”, “2”, “3”, “4”, and “5” are shown in Table 4 (Section 2.3).

**Table 8 ijerph-17-05198-t008:** Health-supportive environment evaluation ^1^.

No.	Name of the Sample Residential Area	Daily Residents’ Necessary Walking Time (min)	Daily Residents’ Triggered Walking Time (min)	Total Walking Time Per Day (min)	Health-Supportive Environment Evaluation (Qualified is “√”, Unqualified is “×”)
01	Yifeng Garden	60	26	86	√
02	Huilongwan Community	13	45	58	×
03	Yajule International Garden	6	30	36	×
04	Jinyang Roman Holiday	13	4	17	×
05	Wanshou Garden	90	60	150	√
06	Changjiang Village Resettlement House	60	13	73	√
07	Kangde Capitol Hill	10	10	20	×
08	Zone A of Haitang xiao yue	19	6	25	×
09	Phase I of Rongqiao City	2	0	2	×
10	Xiangmi Mountain on Rongqiao Peninsula	1	10	11	×

^1^ The data of the frequency of residents’ walking trips and the duration of necessary residents’ walking trips are calculated according to the minimum value.

**Table 9 ijerph-17-05198-t009:** Comparison of model prediction data and real data.

Probability	Y1 Probability (1 to 2 Times Per Week—Value “1”, 3 to 6 Times Per Week—Value “2”, 1 Time Per Day—Value “3”, 2 Times Per Day and above—Value “4”)	Y2 Probability (0 to 5 min—Value “1”, 5 to 15 min—Value “2”, 15 to 30 min—Value “3”, 30 to 45 Min—Value “4”, 45 to 60 min—Value “5” 60 min and above—Value “6”)	Y3 Probability (1 to 2 Times Per Week—Value “1”, 3 to 6 Times Per Week—Value “2”, 1 Time Per Day—Value “3”, 2 Times Per Day and above—Value “4”)	Y4 Probability (0 to 5 min—Value “1”, 5 to 15 min—Value “2”, 15 to 30 min—Value “3”, 30 to 45 Min—Value “4”, 45 to 60 min—Value “5” 60 min and above—Value “6”)
Dependent variable probability “P”	P (Y1 = “1”) = 0.144 P (Y1 = “2”) = 0.686 P (Y1 = “3”) = 0.129 P (Y1 = “4”) = 0.042	P (Y2 = “2”) = 0.016 P (Y2 = “3”) = 0.323 P (Y2 = “4”) = 0.558 P (Y2 = 5) = 0.075 P (Y2 = 6) = 0.028	P (Y3 = “1”) = 0.487 P (Y3 = “2”) = 0.450 P (Y3 = “3”) = 0.056 P (Y3 = “4”) = 0.007	P (Y4 = “1”) = 0.002 P (Y4 = “2”) = 0.020 P (Y4 = “3”) = 0.168 P (Y4 = “4”) = 0.568 P (Y4 = 5) = 0.241
Forecast data	Y1 = “2”	Y2 = “4”	Y3 = “1”	Y4 = “4”
Real data	Y1 = “2”	Y2 = “4”	Y3 = “1”	Y4 = “4”

**Table 10 ijerph-17-05198-t010:** Data adjustment of residents’ walking trips ^1^.

Residents’ Walking Trips Classification	Necessary Residents’ Walking Trips Frequency (Y1)	Duration of Necessary Residents’ Walking Trips (Y2)	Triggered Residents’ Walking Trips Frequency (Y3)	Duration of Triggered Residents’ Walking Trips (Y4)	Walking Trips’ Time Every Day
Value before transformation	“2”	“4”	“1”	“4”	17 min
Data before transformation	3/7 (Times/day)	30 (min/time)	1/7 (Times/day)	30 (min/time)	
Value after transformation	“3”	“4”	“3”	“4”	60 min
Data after transformation	1 (Times/day)	30 (min/time)	1 (Times/day)	30 (min/time)	

^1^ The data of the frequency of residents’ walking trips and the duration of necessary residents’ walking trips are calculated and adjusted according to the minimum value.

**Table 11 ijerph-17-05198-t011:** Adjustment of spatial elements.

Adjustments of Spatial Elements (X) Corresponding to the Frequency of Necessary Residents’ Walking Trips (Y1)			Original Data	Modified Data
Independent variable (X)	XB1	Density of pedestrian access (Pcs/Hm^2^)	0.794	1.389
	XB11	Density of bus routes (Pcs/km)	2.813	5.625
	XC3	Average pedestrian access distance (m)	237.030	217.278
Dependent variable (Y1)	Probability of dependent variable		P (Y1 = “1”) = 0.144 P (Y1 = “2”) = 0.686 P (Y1 = “3”) = 0.129 P (Y1 = “4”) = 0.042	P (Y1 = “1”) = 0.02 P (Y1 = “2”) = 0.35 P (Y1 = “3”) = 0.36 P (Y1 = “4”) = 0.26
	Results of data		Y1 = “2”	Y1 = “3”
**Adjustments of spatial elements (X) corresponding to the frequency of triggered residents’ walking trips (Y3)**			**Original data**	**Modified data**
Independent variable (X)	XD1	Square area within a 500 m walking distance (Hm^2^)	0.200	0.400
	XD2	Distance to the nearest garden (m)	1357.000	700.000
Dependent variable (Y3)	Probability of dependent variable		P (Y3 = “1”) = 0.487 P (Y3 = “2”) = 0.450 P (Y3 = “3”) = 0.056 P (Y3 = “4”) = 0.007	P (Y3 = “1”) = 0.050 P (Y3 = “2”) = 0.401 P (Y3 = “3”) = 0.431 P (Y3 = “4”) = 0.118
	Data results		Y3 = “1”	Y3 = “3”

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
