# Peer review of "The Study on Spatial Elements of Health-Supportive Environment in Residential Streets Promoting Residents’ Walking Trips"

_ijerph, 2020, doi:10.3390/ijerph17145198_

Round 1

Reviewer 1 Report

No more comments.

Reviewer 2 Report

Thank you for a much improved version of the paper.  There are still quite a few areas for revision/clarification.  This time, I am attaching your pdf with comments from me posted as sticky notes.

Author Response

This manuscript is a resubmission of an earlier submission. The following is a list of the peer review reports and author responses from that submission.

Round 1

Reviewer 1 Report

This paper was interesting to read, but I find that there is not much that is new in here.  Some detailed comments are as follows:

  • Methodologically, I think this is fine, although you need to explain the methods a bit more clearly.  For example, what are pedestrian access points exactly?  How many?
  • You mention questionnaires but did you actually survey people?  What was the N?
  • COVID-19 is used unnecessarily as there is no further discussion about it apart from the Intro and Conclusion
  • The variables identified in the paper have been discussed as qualities that afford walkability in research done before this.  You just link it to health but that link is not strong either.  So what else is this paper adding to the knowledge base?  

Author Response

Response to Reviewer 1 Comments

Point 1: Methodologically, I think this is fine, although you need to explain the methods a bit more clearly.  For example, what are pedestrian access points exactly?  How many?

Response 1: We added Table 5 (Page 13, line 240-241) and three footnotes (Page 10-11, Footnotes 3-5) to explain the methods more clearly. The number of entrances and exits in each residential area is different, and the greater the number of residential streets per unit area, the stronger the pedestrian accessibility. The factors related to the number of community entrances and exits mainly include “density of pedestrian access points in the residential community” and “average distance to pedestrian access points” (Tables 4 and 5). And Line 109-111, “According to the questionnaire survey of 10 sample communities, the time and frequency of pedestrian travel were not only limited by the street space, but also affected by the residents' own time arrangement” was added to expound on the method. In addition, we have added supplemental file to supplement the analysis data and process.       

Point 2: You mention questionnaires but did you actually survey people?  What was the N?

Response 2: We expounded on the methods used to collect data (questionnaire and interview) and added the source of data, quantity of data, and logic of data analysis (Page 3, line 86-99):” The distribution of questionnaires in each sample residential area was controlled at 200-300. A total of 2736 questionnaires were issued, of which 2703 were valid questionnaires. We interviewed 30-40 residents of each sample residential area, which totaled 375 participants.” In addition, we have added supplemental file to supplement the analysis data and process.

Point 3: COVID-19 is used unnecessarily as there is no further discussion about it apart from the Intro and Conclusion

Response 3: We deleted the discussion about the coronavirus disease (Intro and Conclusion).

Point 4: The variables identified in the paper have been discussed as qualities that afford walkability in research done before this.  You just link it to health but that link is not strong either.  So what else is this paper adding to the knowledge base?  

Response 4: We added information about how improving the walking environment can promote the health of residents(lines 35-37). The innovation of this paper is to build a space element system of a health-supporting environment based on the residents' health needs. The spatial factors affecting residents' health are determined through interviews and literature review and then combined with the objective real-time data of residents' walking trip. The spatial factor system of the health-supporting environment is obtained through near-term competition. The emphasis is on residents as the center, combining theory with practice, and two dimensions to screen spatial elements.

Other changes:

Page 1 and 4, two footnotes (Footnotes 1 and 2) were added to explain the methods clearly.

Line 69-73, “the introduction about Nan’an District” was added to make the research object clearer.

Line 130-131, “The questionnaire survey of residents' health needs and related literature review can divide the street health supporting environment into the following four aspects:” was added to make the logic of the article clear.

Line 290-291, “References 1” was added to clarify the relationship between research objects.

Reviewer 2 Report

Page 1, line 35-41: it is not necessary to speak too much about the coronavirus disease. In contrast, the authors should introduce something more about what has already been known and what is the evidence gap regarding the aims of the current study.

Page 2, line 50-73: I suggest these paragraphs, that is a conceptual analysis of health-supportive environments, be simplified and integrated into the “Introduction” section.

Page 2, line 74 – page 5, line 106: If these are how you defined and assessed spatial elements of the health-supportive environment, please move them to the section of “Study design and Methods”. You may introduce some detailed concepts in a supplemental file.

Page 5, line 109: Please give a brief introduction about Nan’an District, relative to Chongqing City, including but not limited to population size, area, how many residential communities in this District, etc.

Page 6, line 119-120: Do not understand what this sentence means? “The temporal characteristics were the evaluation criteria for residents’ health......”

Page 6, line 122-124: The authors mentioned several methods to collect data. However, I cannot find a more detailed introduction about how they conducted for each of them. For example, how many interviews with questionnaires were conducted? How were the participants selected? How were the on-site photos analyzed and translated into information? Even in the Result section, many coefficients were reported, but I do not know what original data their analysis was based on.

Author Response

Response to Reviewer 2 Comments

Point 1: Page 1, line 35-41: it is not necessary to speak too much about the coronavirus disease. In contrast, the authors should introduce something more about what has already been known and what is the evidence gap regarding the aims of the current study.

Response 1: We deleted the discussion about the coronavirus disease and added a footnote(Page 1, Footnote 1)and a “References 1”(Page 15, line 290-291) which information about how improving the walking environment can promote the health of residents (Page 1, lines 35-37).

Point 2: Page 2, line 50-73: I suggest these paragraphs, that is a conceptual analysis of health-supportive environments, be simplified and integrated into the “Introduction” section.

Response 2: We move lines 50-73 into the “Introduction” section (Page 2, lines 45-66).

Point 3: Page 2, line 74 – page 5, line 106: If these are how you defined and assessed spatial elements of the health-supportive environment, please move them to the section of “Study design and Methods”. You may introduce some detailed concepts in a supplemental file.

Response 3: We moved Page 2, line 74 – page 5, line 106 to the “Study design and Methods” section. And added “The questionnaire survey of residents' health needs and related literature review can divide the street health supporting environment into the following four aspects(Page 6, line 130-131):” . We also defined spatial elements of the health-supportive environment on page 1, lines 35-37 and explained how we assessed spatial elements of the health-supportive environment on the three footnotes (Page 11-12, Footnote 3-5).

Point 4 : Page 5, line 109: Please give a brief introduction about Nan’an District, relative to Chongqing City, including but not limited to population size, area, how many residential communities in this District, etc.

Response 4: We added information about the Nan’an District (Page 2, lines 69-73).

Point 5: Page 6, line 119-120: Do not understand what this sentence means? “The temporal characteristics were the evaluation criteria for residents’ health......”

Response 5: We provided further explanation of that sentence, interpreted as: ”Length and regularity of daily walking travel time of residents” (Page 3, lines 85),and we were added the footnote (Page 4, Footnote2) to illustrate.

Point 6: Page 6, line 122-124: The authors mentioned several methods to collect data. However, I cannot find a more detailed introduction about how they conducted for each of them. For example, how many interviews with questionnaires were conducted? How were the participants selected? How were the on-site photos analyzed and translated into information? Even in the Result section, many coefficients were reported, but I do not know what original data their analysis was based on.

Response 6: We expounded on the methods used to collect data (questionnaire and interview) and added the source of data, quantity of data, and logic of data analysis (Page 3, line 86-99): ” The distribution of questionnaires in each sample residential area was controlled at 200-300. A total of 2736 questionnaires were issued, of which 2703 were valid questionnaires. We interviewed 30-40 residents of each sample residential area, which totaled 375 participants.” Table 5 (Page 13, line 240-241) and three footnotes (Page 10-11, Footnote 3-5) to explain the methods more clearly. In addition, we have added supplemental file to supplement the analysis data and process.

Other changes:

Line 109-111, “According to the questionnaire survey of 10 sample communities, the time and frequency of pedestrian travel were not only limited by the street space, but also affected by the residents' own time arrangement.” was added to explain the methods clearly.            

Round 2

Reviewer 1 Report

Thank you for the improved version of the paper.  I feel that there would still need to be some revisions to make it publishable.  These comments are highlighted in the attached pdf and comments inserted as 'sticky notes' for you to look at.

One of my main comments deals with the health aspect.  Did you ask your respondents about their health status?  Did you ask them what they would like or need in order to be able to walk more or what is deterring them from walking more currently?  

From what is done now, I feel like an assessment (and good one at that) is just been done of the walkable environments in your study area.  If the connection to health is just by walking more, it is still not clear how you assess the residents' needs/wants if you didn't ask them this.  If you have asked them, pls mention it in your write-up to clarify. If you haven't done this, you should acknowledge these in your limitations.

As mentioned above, other comments are inserted on your pdf file that is attached.

Reviewer 2 Report

  • When was the residential questionnaire survey conducted? How were the 200-300 participants of each residential area selected, by random sampling according to the residential registry or by stopping people on the street? What were the sociodemographic characteristics of the participants? Also, there are similar questions for the 375 participants. How were their collected information analyzed?
  • Please attach the questionnaire that the survey used.
  • What is “walking health needs”, “perspective of residents’ health needs”? Which variables were referring to them?
  • For the value assignment in table 1, is it appropriate for these variables to be transformed into the ordinal variable? They are not equal intervals (distances between adjacent answer categories are equal).
  • The variables in Table 2 and data in the supplemental file were not collected through questionnaire interviews for participants, right? So, how these variables were collected needs to be clarified in the paper.
  • All tables in the supplemental file need to be referred to in the text, if the authors indeed want to present them.
  • How were the residential questionnaire survey and the spatial elements of the residential streets connected and analyzed? The unit of analysis was individual participant (n=2703) or community (n=10)? The 200-300 participants in each community have same spatial characteristics of the health-supportive environment (e.g., area, density, number, etc).
  • The paper needs to be carefully revised for the English language.
